# Bats experience age-related hearing loss (presbycusis)

Yifat Chaya Tarnovsky[1,2], Shahar Taiber[1,5] , Yomiran Nissan[2] , Arjan Boonman[2], Yaniv Assaf[1,3], Gerald S Wilkinson[6] , Karen B Avraham[3,5], Yossi Yovel[2,3,4]

**Hearing loss is a hallmark of aging, typically initially affecting the higher frequencies. In echolocating bats, the ability to discern high frequencies is essential. However, nothing is known about age-related hearing loss in bats, and they are often assumed to be immune to it. We tested the hearing of 47 wild Egyptian fruit bats by recording their auditory brainstem response and cochlear microphonics, and we also assessed the cochlear histology in four of these bats. We used the bats' DNA methylation profile to evaluate their age and found that bats exhibit age-related hearing loss, with more prominent deterioration at the higher frequencies. The rate of the deterioration was ~1 dB per year, comparable to the hearing loss observed in humans. Assessing the noise in the fruit bat roost revealed that these bats are exposed to continuous immense noise—mostly of social vocalizations—supporting the assumption that bats might be partially resistant to loud noise. Thus, in contrast to previous assumptions, our results suggest that bats constitute a model animal for the study of age-related hearing loss.**

## Introduction

Population aging in humans is an international concern (Sanderson & Scherbov, 2010). Aging is the driving factor of various disorders such as neurodegenerative and cardiovascular diseases, cancer, and metabolic diseases (Booth & Brunet, 2016; Hemagirri & Sasidharan, 2022). As the proportion of the elderly population increases around the globe, the need to research and develop strategies to promote healthy aging is becoming ever more critical (Aging Atlas Consortium, 2021).

Bats, which share brain structures with humans and other mammals (Vernes, 2017), offer an emerging model system for aging research because of their extremely long lifespan relative to size, with many bat species living up to at least 40 yr (Pollard et al, 2019): a 6 gr *Myotis brandtii* bat was caught in nature 41 yr after having been first ringed (Podlutsky et al, 2005), whereas in comparison, the

lifespan of a (twice larger) mouse is ~2.5 yr (Liberman, 2020). Consequently, bats might be able to provide us with an important insight regarding the processes and mechanisms of aging (Brunet-Rossinni & Austad, 2004). Indeed, several recent studies have revealed new insights into the mechanisms behind slow aging in bats. These studies suggested that a combination of adaptations, such as hibernation, a low reproductive rate, and cave roosting, probably serves to prolong the lifespan in Brandt's bat (Seim et al, 2013). DNA damage response genes that affect DNA repair and telomere maintenance have been suggested to contribute to the evolution of exceptional longevity in *Myotis* species (Foley et al, 2018). Moreover, bats have developed unique adaptations that counteract inflammation, enabling them to coexist with certain viruses (Gorbunova et al, 2020).

Age-related hearing loss or presbycusis is a hallmark of aging and has been described in multiple species, including mice, humans (Kujoth et al, 2005), gerbils (Gates & Mills, 2005), rats, cats, and primates (Langemann et al, 1999). The neural processing of sound relies on a complex interplay of excitatory and inhibitory interactions (Parthasarathy & Kujawa, 2018; Mollaei et al, 2022), with age-related hearing loss usually arising from irreversible damage in the inner ear, specifically the cochlea, where sound is transduced into electrical signals (Wu et al, 2020). The cochlea is organized tonotopically, maximally responsive to high frequencies at the basal end and low frequencies at the apical end (Anderson et al, 2018). Aging in mammals is accompanied by a progressive deterioration of hearing that usually begins at high frequencies (Huang & Tang, 2010; Krumm et al, 2017) and then spreads toward the low-frequency regions (Wang & Puel, 2020).

As nocturnal mammals, most bat species echolocate by emitting high-frequency ultrasonic signals and processing these signals' returned reflections of obstacles and targets in their environment (Surlykke et al, 2014; Fenton et al, 2016). Thus, although high-frequency hearing confers a survival benefit for many animals, it is essential for the survival of echolocating bats, which rely on it for orienting in their environment (Mao et al, 2017). This combination of extreme longevity and reliance on high-frequency hearing makes bats very relevant models for studying age-related hearing loss. Such research is rare, however, and because of this lack of data, it

---

[1]School of Neurobiology, Biochemistry, and Biophysics, Faculty of Life Sciences, Tel Aviv University, Tel Aviv, Israel   [2]School of Zoology, Faculty of Life Sciences, Tel Aviv University, Tel Aviv, Israel   [3]Sagol School of Neuroscience, Tel Aviv University, Tel Aviv, Israel   [4]School of Mechanical Engineering, Faculty of Engineering, Tel Aviv University, Tel Aviv, Israel   [5]Department of Human Molecular Genetics and Biochemistry, Faculty of Medicine, Tel Aviv University, Tel Aviv, Israel   [6]Department of Biology, University of Maryland, College Park, MD, USA

Correspondence: yossiyovel@gmail.com

has generally been presumed that bats possess resistance to age-related hearing damage (Brunet-Rossinni & Wilkinson, 2009; Peterson, 2020).

Bats have also been suggested to possess mechanisms that protect their hearing from the high-intensity sounds to which they are exposed (Kick & Simmons, 1984; Simmons et al, 2016; Liu et al, 2021). One such contributing factor is the middle ear muscle reflex, which operates to attenuate the amplitude in the bat's inner ear (Henson, 1965; Simmons et al, 2015). Detection thresholds in the big brown bat (*Eptesicus fuscus*) measured after exposure to noise did not vary significantly from pre-exposure thresholds or from thresholds in control conditions (Simmons et al, 2016), and the bats remained able to perform difficult echolocation tasks (Hom et al, 2016). Middle ear muscle reflex protection, however, might not always be sufficient, as discussed by Pilz et al (1997).

One recent study that presented anecdotal evidence of hearing loss in echolocating bats has been suggested as likely because of a combination of stressors (Weinberg et al, 2021), but no study has so far systematically examined the effect of age on hearing in bats.

One of the reasons for this lack of study might be the fact that it has been almost impossible to accurately determine the age of wild bats. Here, we overcame this challenge using a recently developed non-invasive DNA methylation (DNAm) technique (Wilkinson et al, 2021). We examined age-related hearing loss in 47 Egyptian fruit bats (*Rousettus aegyptiacus*) caught in the wild. These bats rely on ultrasonic echolocation (with a peak frequency of ~30 kHz; Yovel et al, 2011) for various orientation tasks even in broad daylight (Eitan et al, 2022), and they have been observed to live for 25 yr (Kulzer, 1979), making them an interesting model for studying age-related hearing loss. In light of the critical importance of hearing for echolocating bats and in light of previous evidence that bats are immune to noise-induced hearing loss, we aimed to test age-related hearing loss in bats, hypothesizing that we will not find any.

Sensorineural hearing loss suggests a pathological condition in either the cochlea or the auditory nerve. The "sensory" component refers to the cochlea, whereas the "neural" component refers to problems primarily in the auditory nerve (Anderson et al, 2018) or secondarily in the auditory relay. Thus, it is important to study changes on either side of the cochlear synapse. To assess hearing sensitivity, we used minimally invasive auditory-evoked brainstem response (ABR) thresholds (Kujoth et al, 2005) recorded in anesthetized bats. ABRs are acoustically evoked electrical responses that can be recorded via intracranial, subdermal, or superficial electrodes. The response consists of a stereotyped waveform generated by synchronous neural activity in the successive early stages of auditory processing, and has proven to be a useful tool for comparing hearing sensitivities across animals (Smotherman & Bakshi, 2019). In addition, we recorded minimally invasive cochlear microphonics (CM), using the same subdermal electrode positioning used for the ABR recordings. The CM response represents activity from the outer hair cells (OHCs), which amplify the sound-induced motions in the inner ear, whereas the inner hair cells translate these motions into the chemical signals that excite the auditory nerve (AN) (Liberman, 2015). To further explore the underlying pathology leading to age-related hearing loss in these bats, cochlear histology was performed in four bats.

ABR recordings revealed a clear age-related hearing loss that is more prominent at the higher frequencies. Both the reduced CM amplitudes with age and a reduced stria vascularis (SV) area with age suggest age-related deterioration in the cochlea. The deterioration in neuronal speed of processing, assessed through a suprathreshold temporal analysis of ABRs, might represent neuronal presbycusis.

Furthermore, to assess the noise that our bats are typically exposed to in their colony we placed several continuous calibrated microphones in their roost. This procedure revealed that bats are routinely exposed to very high broadband noise levels, which are known to have a negative effect on the auditory system of mammals. Taken together, our results provide the first evidence that bats are susceptible to age-related hearing loss, but show that hearing loss in bats is similar to that observed in humans despite routine exposure to very loud noise.

# Results

## ABRs and the bat audiogram

To assess bat hearing thresholds, we played 1-ms tones of 35, 30, 24, 18, 12, and 6 kHz and a 0.1-ms click signal (with most energy up to 10 kHz). Playback intensities spanned from at least 10 dB below to 10 dB above the suspected threshold for each frequency, but not lower than 10 dB sound pressure level (SPL; whenever we use the term dB SPL, it is always relative to 20 $\mu$Pa) for pure tones (and 20 dB SPL for clicks), and not higher than 90 dB SPL. The hearing threshold at each frequency was defined offline as the lowest intensity at which the peak-to-peak amplitude of the response exceeded a 6.5*SD criterion (Fig 1A; see the Materials and Methods section).

We used the thresholds obtained from the ABRs to reconstruct the bats' hearing audiogram for two age groups. Threshold differences between the youngest bats (evaluated age of the group is 2.79 ± 0.3 yr) and the oldest bats (evaluated age of the group is 11.99 ± 1.12 yr) are up to ~10 dB (Fig 1B). The mean thresholds for the entire group (Fig 1D) agreed with previous ABR-based audiograms recorded from *Rousettus* (Belknap & Suthers, 1982), with the lowest (best) threshold at 12 kHz (~23 ± 10 dB SPL) and a threshold increase of ~24 dB SPL at 35 kHz (with averaged threshold of 47 ± 12.5 dB SPL).

Next, we analyzed the effect of age and sex on hearing thresholds. Sex differences were insignificant, whereas age differences were significant for five of the six frequencies and revealed elevated hearing thresholds in older animals (Fig 1D, $P$ = 0.018, 0.099, 0.026, 0.037, 0.003, and 0.009 for the frequencies 6, 12, 18, 24, 30, and 35 kHz, respectively, for each frequency, a generalized linear model (GLM) with the hearing threshold set as the explained parameter and age and sex as fixed factors; see the Materials and Methods section).

Moreover, hearing deterioration was significantly faster for the higher frequencies, with the age-related rate (slope) significantly higher at 35 kHz (1.44 ± 0.53 dB threshold elevation per year) and 30 kHz (1.31 ± 0.42 dB threshold elevation per year) than at 24 kHz (0.95 ± 0.44 dB threshold elevation per year), 18 kHz (0.68 ± 0.3 dB threshold elevation per year), 12 kHz (0.77 ± 0.45 dB threshold elevation per year), and 6 kHz (0.87 ± 0.35 threshold elevation per year) ($P$ = 0.0001 for the comparison between 24 and 30 kHz, and

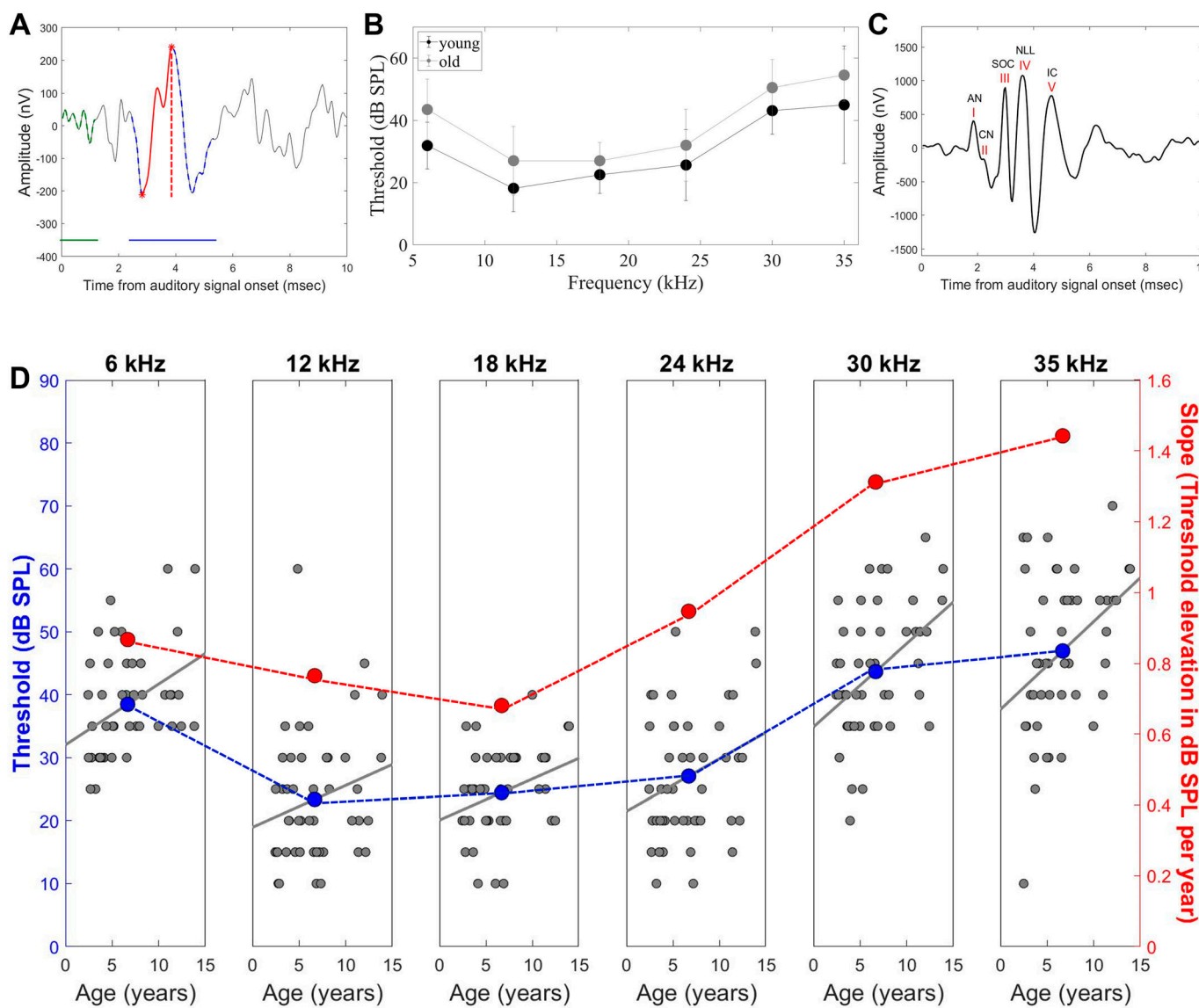

**Figure 1. Bats exhibit age-related hearing loss.**
**(A)** 10-ms averaged response window recorded in response to a 45 dB SPL 30-kHz playback, for example. The average of 512 responses is shown for one bat. The green window (horizontal line) was used for calculating noise SD; the blue horizontal line depicts the search window where the minimum peak of the response was detected; and the red vertical dashed line depicts the absolute peak-to-peak amplitude used to define the hearing threshold. **(B)** Audiogram for two age groups. Black—eight youngest bats (with age that is smaller than the mean – SD of all ages. Evaluated age of the group is 2.79 ± 0.3 yr). Gray—10 oldest bats (with age that is larger than the mean + SD of all ages. Evaluated age of the group is 11.99 ± 1.12 yr). Threshold differences between the groups are up to ~10 dB. **(C)** Averaged ABR waveform to a 0.1-ms click signal played by the speaker in alternating polarity at a suprathreshold intensity of 30 dB sensation level. The average of 512 responses is shown for the same individual. It is commonly accepted that wave I represents the compound response from the auditory nerve (AN), whereas the later waves represent responses from the ascending auditory pathway: the cochlear nucleus (CN), the superior olivary complex (SOC), the nucleus of the lateral lemniscus (NLL), and the inferior colliculus (IC). **(D)** Audiogram results. Different frequencies are displayed from low to high (6, 12, 18, 24, 30, and 35 kHz). Gray points—individual thresholds as a function of age for each frequency. Each data point represents a single individual, and the gray line shows the linear fit ($P$ = 0.018, 0.099, 0.026, 0.037, 0.003, and 0.009 for the frequencies 6, 12, 18, 24, 30, and 35 kHz, respectively). Blue points show the average threshold in dB SPL for each frequency. Red points—the mean regression slope of the gray line depicting the threshold elevation (in dB per year) for each frequency (n = 46).

$P < 0.0001$ for the rest of the comparisons; slope comparisons between the different frequencies were analyzed using a two-tailed $t$ test with a Bonferroni correction for multiple comparisons). In addition, the slope at 24 kHz (0.95 ± 0.44 dB threshold elevation per year) was higher than at 18 kHz (0.68 ± 0.3 dB threshold elevation per year) ($P$ = 0.001) (Fig 1D).

**Cochlear microphonics (CM)**

For further verification of the mechanism underlying hearing deterioration, minimally invasive CM were recorded in response to a 90 dB SPL 0.1-ms click signal that was played at opposite polarities (condensation and rarefaction; Fig 2A).

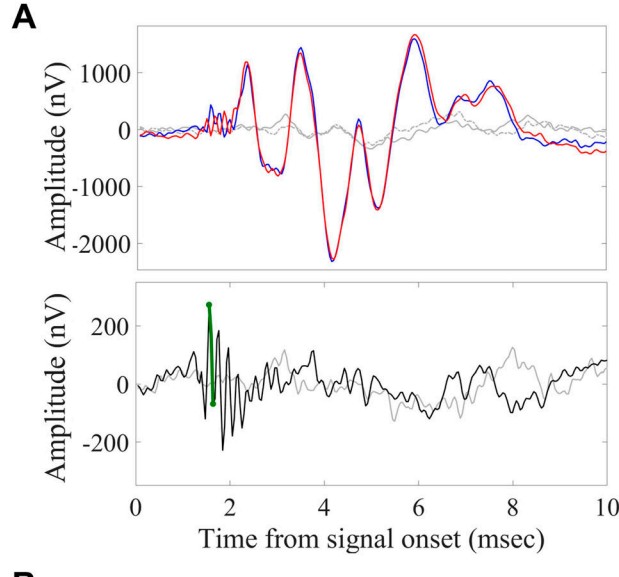

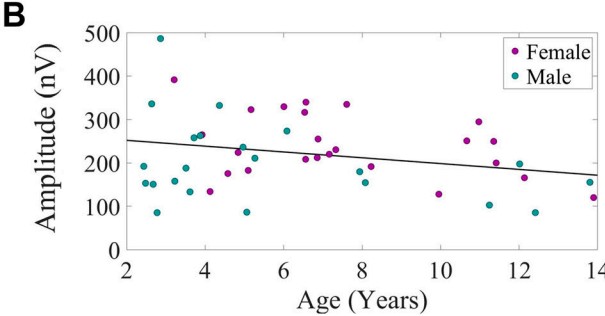

**Figure 2.   Bats exhibit an age-related decrease in cochlear microphonics (CM) response.**
**(A)** Top—the averaged EEG response to a 90 dB SPL 0.1-ms click signal played to the right ear via an ~30-cm ear tube, shown separately for the two opposite polarities (blue: condensation; red: rarefaction). Gray lines show the noise floor (solid: condensation; dashed: rarefaction), collected by playing the signal through a blocked tube. Each response shows the average of 2,000 clicks played to one bat. Bottom—the CM response (green) extracted by subtracting the response to rarefaction from the response to condensation (black), and the noise floor (gray). **(B)** Effects of age and sex on CM amplitude. Each point represents one individual. Black—age regression line, n = 46.

Analysis of the effect of age and sex on the CM response (Fig 2B) revealed a significant reduction in CM amplitude with age, and a significant difference between the sexes, with higher amplitudes (i.e., better sensitivity) in females compared with males ($P$ = 0.016 and $P$ = 0.027, respectively, GLM with the CM amplitude set as the explained parameter and age and sex as fixed factors; see the Materials and Methods section).

## Histology

To further evaluate cochlear health, cochlear histology was performed for four female bats (Fig 3A). The bats' inner ears were dissected and fixed in 4% PFA, and paraffin-embedded tissue blocks were sectioned at the mid-turn of the cochlea and stained with hematoxylin and eosin. The stria vascularis (SV) is the lateral wall ion transport system of the cochlea, and its function is crucial

for normal hearing (Gu et al, 2021). Previous studies have found that the SV cross-sectional area shrinks with age and is correlated to auditory functioning, suggesting a role for SV deterioration in age-related hearing loss (Schulte & Schmiedt, 1992; Hequembourg & Liberman, 2001). Indeed, the SV cross-sectional area was found to decrease with age (Fig 3B).

The hearing thresholds of the bats correlated with their SV area, showing higher (worse) thresholds with smaller SV areas for most of the frequencies, and significantly so for 18 kHz (Fig 3C) ($P$ = 0.026, GLM with the hearing thresholds (dB SPL) set as the explained parameter and the SV area as a fixed factor).

### Neuronal processing

Aging can result in a significant deterioration of signal transmission in the auditory nerve and brainstem (Pürner et al, 2022). To gain more insight into a potential neuronal age-related hearing loss, we examined the waveform of ABRs recorded in response to supra-threshold intensity clicks (Parthasarathy & Kujawa, 2018). These waveforms were stable and contained several identifiable peaks within a 5-ms time frame after the onset of the playback signal. These waves are known to represent the afferent neuronal path along the auditory nerve and up to the inferior colliculus (IC) (Fig 1C).

Thresholds for the 0.1-ms click signal were obtained similar to those for the pure tones, with several modifications, as detailed in the Materials and Methods section. These thresholds also showed an elevation with age ($P$ = 0.043, GLM with the hearing threshold set as the explained parameter and age and sex as fixed factors) (Fig 4A).

We measured the latencies of waves 1, 4, and 5 of each bat's 30 dB suprathreshold response, which were easier to detect across individuals, and the inter-peak interval (IPI) between wave 1 and wave 4 (see the Materials and Methods section).

Our suprathreshold temporal analysis revealed a significant prolongation in the latency of wave 4 and a significant prolonged IPI between wave 1 and wave 4 ($P$ = 0.031 and $P$ = 0.005, respectively, GLM with the latency or IPI set as the explained parameter and age and sex as fixed factors), which represents activity from cranial nerve VIII until the nucleus of the lateral lemniscus (NLL) (Starr, 1976). The large individual variability observed in the response latency of wave 5 might reflect the complexity of the neural circuitry (Simmons et al, 2022) at the inferior colliculus level (Fig 4B).

### Noise exposure evaluation

Sound recordings in a fruit bat cave revealed almost non-stop exposure to loud conspecific vocalizations. We estimated the noise intensities a bat is exposed to in the roost at two key frequencies (6 and 32 kHz) at windows of 30 s (we excluded 6 h per night when most of the bats were outside the cave; see the Materials and Methods section and Fig 5A). Average noise levels at 32 kHz were ~28 dB, and the maximum levels (peak-to-peak) exceeded 75 dB SPL in ~44.34% of the recordings and 85 dB SPL in ~2.94% of the recordings, whereas the average noise levels at 6 kHz were ~52.5 dB SPL, and the maximum levels (peak-to-peak) exceeded 90 dB SPL in ~63.57% of the recordings and 100 dB SPL in ~10.54% of the

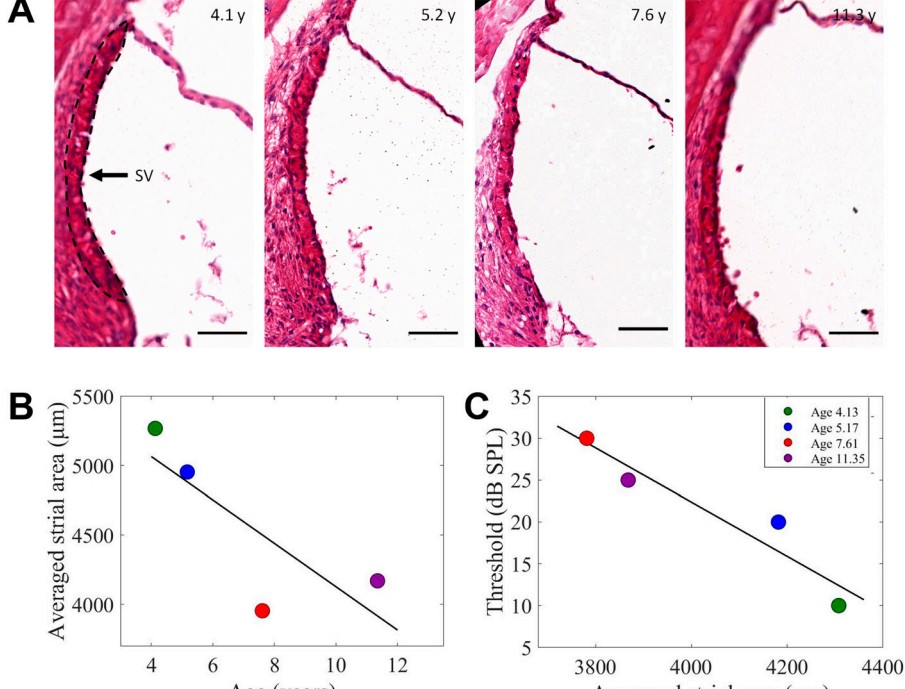

**Figure 3. Bats exhibit an age-related decrease in the stria vascularis (SV) area that is correlated with hearing threshold elevation with age.**
**(A)** Hematoxylin and eosin (H&E) staining of the cochlea in four bats. The SV cross-sectional area is depicted in a dashed line. **(B)** Averaged strial area as a function of age. **(C)** Hearing threshold at 18 kHz as a function of the averaged strial area.

recordings. We placed our microphones at a distance of ~10 cm from the nearest bat, whereas the bats in the colony are tightly clustered (touching each other), and thus, they are probably exposed to even louder noise. The analysis of the entire frequency spectrum of the environmental noise exposure of the bats was done using recordings from the laboratory (Fig S1). The results revealed an astonishing ~140 dB SPL maximum levels of vocalizations at 12 kHz, estimated at a distance of 10 cm from the bats. This frequency, of the six tested in the audiogram, is the only one that did not show a statistically significant deterioration with age. Moreover, a significant inverse correlation was found between the rate of age-related hearing loss and the noise intensity per frequency, with slower deterioration seen for frequencies with larger exposure to intense conspecific noise (Fig 5B, $P = 0.004$, GLM with the hearing deterioration (dB SPL per year) set as the explained parameter and the noise level as a fixed factor).

## Discussion

It has been commonly assumed that bats are exceptionally resilient to noise and age-related hearing damage (Brunet-Rossinni & Wilkinson, 2009; Peterson, 2020). Here, we show for the first time that bats (the Egyptian fruit bat, *Rousettus aegyptiacus*) too experience age-related hearing loss or presbycusis, with a sharper deterioration at the higher frequencies. This represents a typical pattern of age-related hearing loss in mammals (Anderson et al, 2018), including humans and mice (Li & Borg, 1991; Keithley, 2020).

Our results of an average 1.44 dB threshold elevation per year at 35 kHz and an average 1.31 dB threshold elevation per year at 30 kHz

are in accordance with the deterioration known in humans, which ranges between 1.2 and 1.5 dB per year for high frequencies (typically up to 8 kHz, Kocher, 2009; Lee et al, 2005; Rigters et al, 2018).

Sensitive hearing, especially at high frequencies, is necessary for echolocating bats (Mao et al, 2017), because they must be able to detect the faint echoes that return immediately after the much louder emitted echolocation signals (Grinnell, 2018). Notably, as bats typically live very long (Pollard et al, 2019), hearing impairment might have a serious effect on their fitness. Indeed, a recent sensorimotor model of bat foraging showed that reducing (improving) the hearing threshold in insectivorous bats significantly improved hunting (Mazar & Yovel, 2020), and thus, an elevation in threshold would impair their foraging. The fruit bats that we studied here rely on echolocation for various tasks (Holland et al, 2004; Yovel et al, 2010), but they also heavily rely on vision when possible (Danilovich & Yovel, 2019; Eitan et al, 2022). It is thus important to replicate our tests in bats with poor vision (Chase, 1981; Heffner et al, 2007) where echolocation is nearly the solely orientation sensory modality. Interestingly, dolphins with high-frequency age-related hearing loss tend to compensate using echolocation clicks with lower center frequencies (Strahan et al, 2020). Although there is currently no evidence of age-related adjustments of echolocation frequencies in bats, such adjustments might assist aging bats.

At the behavioral level, age-related hearing loss might be associated with exposure to noise. Aging is a complex process involving many systems (Aging Atlas Consortium, 2021), but some evidence suggests that presbycusis results not only from the aging process per se, but also from accumulated exposure to environmental noise over the lifetime (Gates & Mills, 2005; Fernandez et al, 2020). Indeed, many bats fly in an ultra-noisy environment where they are exposed to both their own and their conspecifics' high-

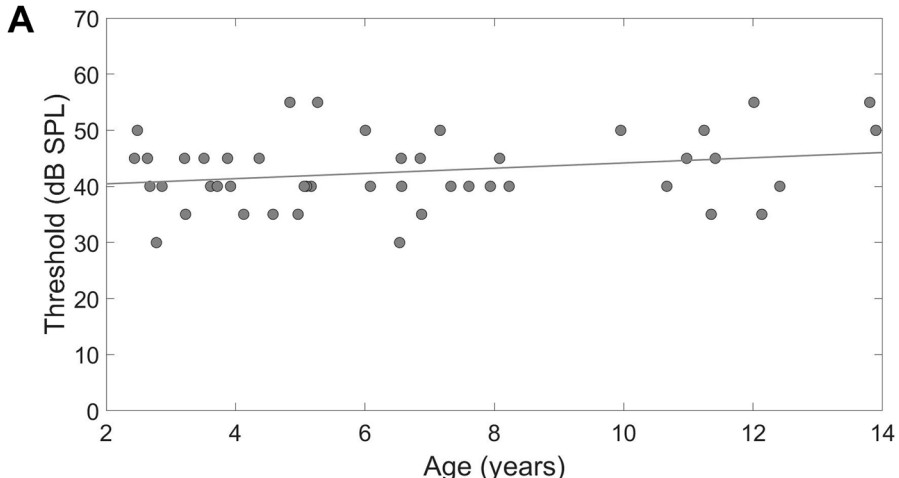

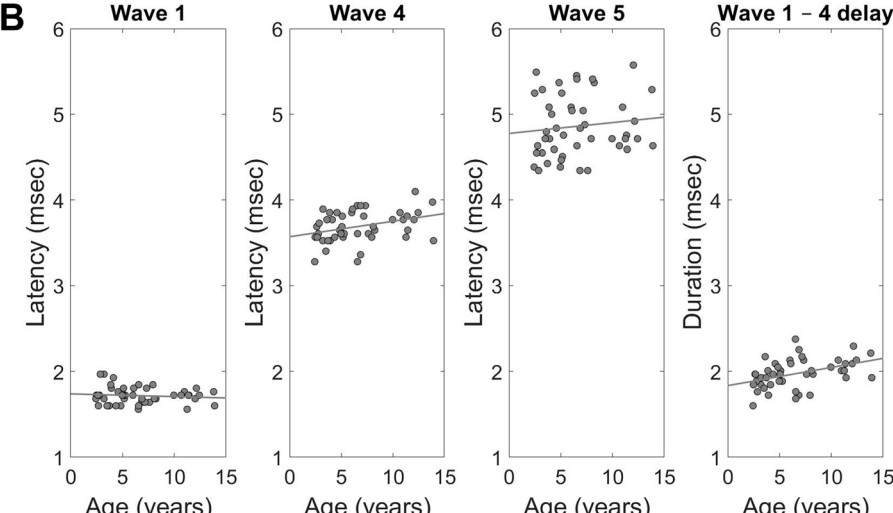

**Figure 4. Bats exhibit an age-related deterioration in neuronal processing.**
**(A)** Individual thresholds for the 0.1-ms click signal and the age-dependent linear regression line. **(B)** From left to right—latencies and age regression lines for waves 1, 4, and 5, and the inter-peak interval between waves 1 and 4 in ms, retrieved from each individual's 30 dB suprathreshold response to the click signal. Each point represents a single bat, n = 46.

frequency echolocation sounds, with SPL intensities that can reach ~120 dB SPL (Surlykke & Kalko, 2008), known to damage the hearing systems in other mammals (Simmons et al, 2018). Moreover, many bat species roost in very noisy colonies with many thousands of individuals (Ducummon, 2000) that live in compact clusters (Rysgaard, 1941; McNab, 1969), where loud social vocalizations (Porter, 1979) that contain low frequencies (Furmankiewicz et al, 2011; Knörnschild, 2014) are continuously emitted. These low-frequency social calls can also be very loud emitted at sound pressure levels of up to 110 dB SPL (Hoffmann et al, 2008). Our noise measurements in the fruit bat roost confirmed these results and also suggest a nearly continuous exposure to noise. Our measurement revealed that the bats in the roost are exposed to social calls at levels that exceed ~100 dB SPL every ~5 min, accounting for more than 100,000 re-current exposures per year.

However, the frequency-dependent hearing loss was not cor-related to the noise. Analyzing the entire frequency spectrum of the conspecific noise revealed an inverse correlation between the rate of age-related hearing loss and noise intensity, with slower de-terioration seen for frequencies with more exposure to vocalization

noise. Specifically, communication vocalizations were louder at frequencies with better hearing thresholds, and these frequencies were less affected by age (Fig S1). Exposure to noise thus does not seem to be the only driver of hearing loss.

In light of this noisy environment, protection of the bat's delicate inner ear appears to be necessary (Henson, 1965), and it has been suggested that bats possess mechanisms that protect their hearing from the high-intensity sounds to which they are exposed (Kick & Simmons, 1984; Liu et al, 2021). Several previous studies suggested that bats might be immune to noise damage, that is, that bat echolocation remains intact (Hom et al, 2016; Simmons et al, 2018), that they do not experience hearing loss (Simmons et al, 2015, 2016), and that their cochlear hair cells remain undamaged (Liu et al, 2021) after noise exposure that causes hearing loss in other mammals. Notably, these studies exposed the bats to noise levels that are up to ~120 dB SPL, but most of them used exposure duration of up to 1 h, making it difficult to conclude whether bats are immune to long exposure to noise.

With sound pressure levels greater than ~100 dB SPL and short duration of less than 1 s, bats' social calls meet the definition of

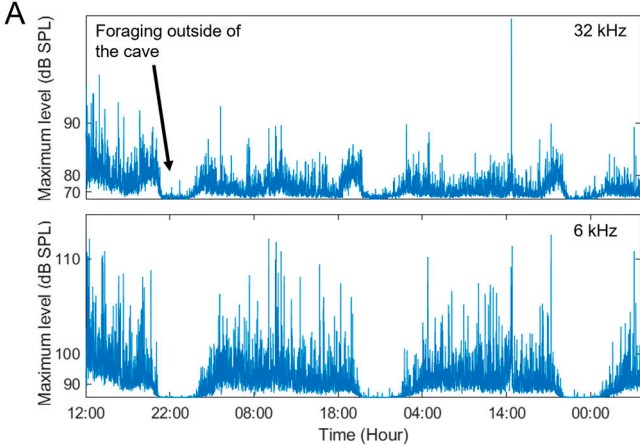

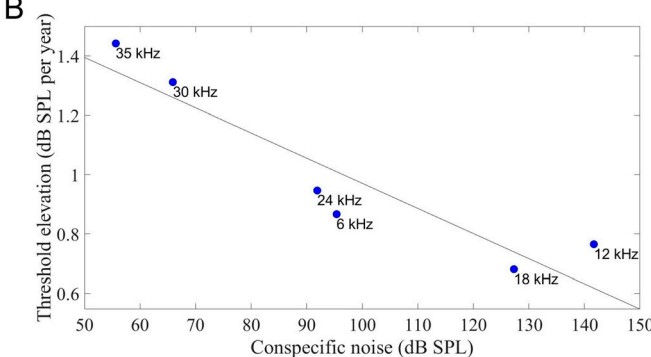

**Figure 5. Noise exposure and the relation to age-related hearing loss.**
**(A)** Maximum noise levels in a roost with thousands of bats. Top—key frequency of ~32 kHz. Bottom—key frequency of ~6 kHz. The microphone was placed at a distance of ~10 cm from the nearest bat. **(B)** Inverse correlation was found between the rate of age-related hearing loss and the maximum intensity of conspecific vocalizations (estimated at a distance of 10 cm from a cluster of 13 bats in the laboratory). The line depicts the linear fit.

exposure to impulse sounds, which are especially hazardous to the ear (Miłoński & Olszewski, 2007).

With a minimal latency delay of ~80 ms between the onset of the sound and the contraction of the middle ear muscles in humans (Boothalingam & Goodman, 2021), both a sudden onset of loud sound and prolonged sound exposures can cause a decrease in the ability of the middle ear muscles to protect the ear from damage (Sherburn et al, 2014).

Although the middle ear muscles of bats contract during self-sound emission and relax afterward to protect the ear from their own loud echolocation signals and to make it sensitive to the perception of weak echoes (Henson, 1965; Suga & Jen, 1975), these muscles would play only a minor role in attenuating the amount of stimulation provided by echolocation sounds emitted by other bats nearby (Suga et al, 1974; Weinberg et al, 2021). Communication sounds with components lower than 20 kHz are, however, longer than 10 ms so that these may be effectively attenuated by the middle ear muscle reflex (Suga & Jen, 1975).

Impulse noise exposures might also cause hidden hearing loss involving cochlear synaptopathy (Qi et al, 2022), which precedes age-related hair cell damage and/or threshold elevation

(Sergeyenko et al, 2013) and can accelerate the cochlear aging (Fernandez et al, 2020) with accumulated repeated noise exposures (Luo et al, 2020). This synaptopathy is associated with a reduction in the amplitude of wave I of the auditory brainstem response (ABR) (Bramhall et al, 2017), which we did not find in the current work.

When taken together, the very high levels of noise that fruit bats are exposed to and the mild (similar to human) levels of age-related hearing loss (i.e., ~1.3 dB per year at the higher frequencies) suggest that they might have some special adaptations to cope with very noisy environments. One such adaptation in addition to their middle ear reflex function might be their high defense against oxidative stress (Chionh et al, 2019; Hanadhita et al, 2019; Lagunas-Rangel, 2020; Irving et al, 2021), which is known to contribute to noise effects on the hearing (Fechter, 2005; Henderson et al, 2006; Yildirim et al, 2007; Umugire et al, 2019).

What is the physiological hearing loss mechanism in fruit bats? We recorded CM as a method to assess OHC functionality (Frost & Olson, 2021). We also used histology to examine the stria vascularis (SV) area in the cochlea. Our findings including reduced CM amplitudes and SV area reduction with age suggest age-related deterioration at the cochlear level.

We also find evidence of hearing loss at the neuronal level. Our suprathreshold temporal analysis revealed a significant prolongation in the latency of wave 4, and a significant prolonged IPI between wave 1 and wave 4, which represents activity from nerve VIII to the nucleus of the lateral lemniscus (NLL) (through the cochlear nucleus [CN] and the superior olivary complex [SOC]) (Starr, 1976). Wave latency seems to increase with age in both humans (Lotfi & Zamiri Abdollahi, 2012) and mice (Kobrina et al, 2020). Some potential causes include alterations in brainstem auditory fibers (Pürner et al, 2022), a decrease in myelin density (Eckert, 2011; Sharma et al, 2016), or an age-related decrease in synaptic function (Deak & Sonntag, 2012).

This deterioration in processing speed is typical of neuronal presbycusis and can result in poorer speech understanding in humans (Gates & Mills, 2005). It might also interfere with bat echolocation ability. To assess target distance, echolocating bats analyze the time intervals between their emitted biosonar pulses and the echoes returning from objects. The time course of acoustic events is thus a critical element for the recognition of meaningful sounds (O'Neill & Suga, 1982).

In summary, the mechanism behind age-related hearing deterioration in bats appears to involve a mixture of pathologies, at the cochlear level and at the neuronal level. How bats deal with hearing loss despite their crucial reliance on hearing for sensing is yet to be revealed.

# Materials and Methods

### Animals and permission

47 adult Egyptian fruit bats (*Rousettus aegyptiacus*; 24 female and 23 male) participated in the ABR and CM recordings. The bats were all adults (older than 1 yr), based on their weight and forearm length. Genomic DNA was extracted from wing samples, and

methylation data (DNAm) were generated. In addition, cochlear histology was assessed in four females of these bats.

Bats were captured in the wild with permission of the Israeli National Park Authority, and all experiments were performed with permission from the Tel Aviv University Institutional Animal Care and Use Committee (permit numbers 04-21-043 and 04-20-023).

## Anesthetization

The bats' weight and forearm measurements were taken, and they were anesthetized by subdermal injection of an anesthetization cocktail comprising 0.48 ml water, 0.1 ml Domitor (1 mg/ml), and 0.05 ml midazolam (5 mg/ml). The bats were injected with a 0.25 ml dose of the cocktail and an additional half-dose if needed. Eye moisture drops were given to prevent eye dehydration.

Anesthesia is known to affect both the central and peripheral nervous systems (Osanai & Tateno, 2016) and has been shown to affect ABRs in mice, rats, birds, and lizards. In frogs, the level of anesthesia affected the amplitude, threshold, and latency of ABR (Cui et al, 2017). In mice, anesthesia with ketamine/xylazine caused a significant prolongation of ABR-peak latencies and inter-peak latencies, and a significant upward shift (8.0 ± 1.8 dB) of ABR thresholds as compared to the awake condition (van Looij et al, 2004). However, anesthetized animals are still widely used in auditory research because of their stability and controllability (Osanai & Tateno, 2016). Research on infants and children found that no significant difference could be observed between the recordings obtained in the awake state or when unconscious (Sohmer et al, 1978; Bocskai et al, 2013), whereas another study even suggested that the large difference in spontaneous EEG activity between awake and sedated children indicates that sedation should be used for estimation of hearing thresholds based on ABR (Knaus et al, 2019).

To date, with the exception of one study (Simmons et al, 2022), ABRs in bats have all been recorded under anesthesia (Hörpel & Firzlaff, 2020; Wetekam et al, 2020; Geipel et al, 2021; Lattenkamp et al, 2021).

## ABR recordings

ABR recordings followed Taiber et al (2021). We used a calibrated setup comprising an RZ6 multiprocessor, an MF1 speaker (Tucker-Davis Technologies), and a calibration microphone (ACO Pacific), all controlled by BioSigRZ software (Tucker-Davis Technologies). Recordings were conducted in an acoustic chamber (MAC-1, Industrial Acoustic Company). Bat body temperature was maintained throughout the experiment using a 37°C heating pad. Auditory signals were played with a speaker placed 10 cm in front of the bat. Responses were picked up by three subdermal electrodes: on the midline forehead, underneath the right ear, and on the left mastoid, and digitized at a sample rate of 24.4 kHz.

Bats were presented with a 0.1-ms click stimulus (with most energy up to 10 kHz), and subsequently with 1-ms tone bursts of 35, 30, 24, 18, 12, and 6 kHz. This set of frequencies encompasses the main hearing range of adult *Rousettus aegyptiacus* (Koay et al, 1998). For each auditory signal type (click or tone bursts) and each intensity tested, signals were played 512 times in alternating polarity (one signal as condensation, the following one as rarefaction, and so on) at a rate of 21 signals per second. The first 10-ms response window starting from the auditory signal onset was collected for each one of the 512 repetitions, and these were averaged to generate the overall 10-ms window response, which was displayed live in the BioSigRZ software. Click signals were presented at constant intensity levels from low to high (20 to 85 dB SPL) in steps of 5 dB, and the tones were tested using a manual approach (Beck et al, 2007). Here, tone signals were presented from 10 dB SPL up to 90 dB SPL in steps of 5 dB. When the waveform in the live presentation seemed to have reached a threshold, two additional intensity levels were recorded (5 and 10 dB above the suspected threshold) before moving to the next frequency. Because for the first ~15 bats, thresholds for some of the frequencies (6, 30, and 35 kHz) were usually higher than 30 dB SPL, recordings at those frequencies for the following bats started at higher intensities than 10 dB SPL (usually not higher than 20 dB SPL). If the threshold was visible in the live presentation at lower intensities than expected, lower intensities were tested too. To summarize, final recordings covered an intensity range of at least two levels (10 dB) below to two levels above the suspected (checked visually from the live presentation) threshold for each frequency, but not lower than 10 dB SPL and not higher than 90 dB SPL.

## ABR analysis

### Audiogram generation

The determination of hearing thresholds involves interpretation of the ABR waveform (Wang et al, 2021). ABR thresholds are generally determined as the lowest sound intensity at which a waveform is manually observed (Sato et al, 2010; Muniak et al, 2018; Buran et al, 2020). Because automatic wave-detection algorithms (Wimalarathna et al, 2021) should improve objectivity and reproducibility, here we added an adaptation of the 3SD/4SD method for determining the ABR thresholds. This method uses the SD of the pre-signal window. A significant response is considered if its peak exceeds 3*SD (Stollman et al, 1996) or 4*SD (Brantberg et al, 1999) of the pre-stimulus voltage variation.

For each bat, an in-house MATLAB script screened the ABR recordings for each of the frequencies (35, 30, 24, 18, 12, and 6 kHz) from the highest to the lowest tested intensity. The ABR system records 10-ms response windows that start with the onset of the auditory signal. The average SD of the first 1 ms from the 10-ms response window was therefore calculated for each intensity. This time window does not include components of the ABR response and should therefore represent the averaged noise. The ABR response to tones was searched in a 3-ms time window between 2.5 and 5.5 ms of the full 10-ms averaged response window. A minimum peak was first detected, and a maximum peak was then detected within a time window of ~1.5 ms following the minimum peak.

The absolute difference between the two peaks was calculated and determined as the peak-to-peak amplitude of the response (Fig 1A). If this amplitude exceeded 6.5-fold the SD (and not threefold or fourfold, because the peak-to-peak amplitude was used here and not the maximum peak), the script moved to test the next, 5 dB lower intensity. If the peak-to-peak amplitude was smaller than 6.5*SD in two consecutive levels, the threshold was

**Life Science Alliance**

determined as the lowest intensity at which the peak-to-peak amplitude exceeded the 6.5*SD criterion, and the script moved on to the next frequency.

Hearing thresholds were obtained for all six frequencies for 46 bats. One bat out of the 47 tested demonstrated residual hearing in the 6-kHz frequency (with a threshold of 70 dB SPL) and lacked response to both the higher frequencies and the click signal (tested here up to 90 dB SPL). Its test was repeated a week later with similar results, and it was removed from further analysis. This individual was not the oldest subject—its estimated age was 7.15 yr; whereas the other bats' ages ranged from 2.44 to 13.9 yr, with a mean of 6.67 ± 3.33 yr. Idiopathic hearing loss in bats was reported recently for the big brown bat (*Eptesicus fuscus*), likely because of a combination of stressors (Weinberg et al, 2021).

## Suprathreshold temporal analysis

The ABR waveform in response to suprathreshold intensity displays several identifiable peaks (Fig 1C). It is commonly accepted that wave I represents the compound response from the auditory nerve, whereas the later waves represent responses from the ascending auditory pathway (Linnenschmidt & Wiegrebe, 2019).

Clicks have been popular stimuli for the study of the ABR, because of their brief duration and broadband spectrum (Burkard & Moss, 1994). Thresholds for the click signal were obtained here similar to pure tones, with a few modifications: the search window was between ~1 and 3 ms, focusing on latencies that represent wave 1 at about threshold intensities. The maximum peak in this time window was searched first, and the minimum peak was searched in a time window of ~1 ms following the maximum peak. The average SD was computed for the last 1 ms from the 10-ms response window in addition to the first 1 ms. If the peak-to-peak amplitude was smaller than 7*SD in two consecutive levels, the threshold was determined as the lowest intensity at which the peak-to-peak amplitude exceeded the 7*SD criterion.

Suprathreshold temporal measurements were taken from the response to the click signal at an adjusted 30 dB SL (sensation level, 30 dB above the threshold response of each bat). Hearing thresholds are known to represent hair cells' function, and the purpose of ABR recordings at a fixed suprathreshold of 30 SL is to eliminate the cochlear state and examine neuronal changes only (Parthasarathy & Kujawa, 2018). The measurements taken were those of the latencies (which are easier to obtain than accurate and reproducible measurements of the amplitudes of evoked potentials [Møller & Jannetta, 1985] of waves 1, 4, and 5, which were more repeatable between individuals (than waves 2 and 3) and the IPI between wave 1 and wave 4, which was the largest wave that was observed and demonstrated less latency variation than wave 5. Wave 4 represents activity from the nucleus of the lateral lemniscus (NLL) (Starr, 1976). A MATLAB script was written to analyze each bat's 30 dB SL response. Wave 1 maximum peak was searched in a time window of 1.5–2 ms. This is earlier than the search window used previously for threshold identification because the response to higher intensities appears earlier (Rouillon et al, 2016). Wave 4's maximum peak was searched in a 1-ms window starting at a latency slightly higher than 3 ms, and wave 5's maximum peak was searched in ~1-ms window starting at wave 4's minimum peak.

## CM recordings

To evaluate the function of the OHC, minimally invasive CM were recorded using the same subdermal electrode positioning used for the ABR recordings. Our method is similar to the one used in humans in which the CM response is recorded via surface electrodes (Sohmer & Pratt, 1976; Rance et al, 1999; Shi et al, 2012). Similar recordings using subcutaneous electrodes were used before on cats (Laukli & Mair, 1983) and Wistar rats (Heidari et al, 2018).

CM were obtained similar to ABRs, with some modifications:

The bats were presented with a 0.1-ms click that was transferred via a 30-cm-long air tube connected on one end to the speaker and serving as an earphone. The other end of the tube was coated with a foam tip and placed at the entrance to the right auditory canal. The intensity at the foam tip end was calibrated to 90 dB SPL, and the signals were played 2,000 times for each of the two polarities separately (condensation and rarefaction). These recordings were then repeated with a similar but blocked tube, in order to rule out possible artifacts of the auditory signal (Neary & Lightfoot, 2012) that are similar to the CM response, which is known to mimic the sound waves.

This control is twofold: first, the CM arrive later in time because of the transmission time in the tube (~1 ms for a 30-cm tube), compared with artifacts that occur immediately upon signal presentation; and second, artifact responses would also be visible in the blocked tube mode when the auditory signal is played but does not reach the subject's ear, whereas real CM response from the cochlea will be present only when the tube mode is open.

Indeed, averaged CM response was ~sixfold larger than the possible artifact (Fig S2) and significantly different (*P* < 0.0001, GLM with the CM amplitude set as the explained parameter and the insert phone tube mode [open or blocked] as a fixed factor, n = 46).

## CM analysis

Periodic sound waves traveling through air consist of alternating regions of compression (i.e., condensation) and decompression (i.e., rarefaction) of air molecules (Skoe & Kraus, 2010). "Condensation" and "rarefaction" are defined according to the initial phase of the auditory signal (Coats et al, 1979).

After the collection of responses to both polarities (condensation and rarefaction), adding the response to rarefaction to the response to condensation ([condensation + rarefaction]/2) will accentuate the lower frequency components of the response—the ABR; whereas subtracting the response to rarefaction from the response to condensation ([condensation—rarefaction]/2) will bias the higher frequency components by maximizing the spectral response—CM (Skoe & Kraus, 2010). Subtraction of the responses to the two polarities was done in MATLAB. The maximum peak and the following minimum peak of the CM were searched in a short window of 0.25 ms starting at a latency of 1.5 ms, and the absolute amplitude between them was calculated. This procedure was carried out for both the open tube mode and the blocked tube mode. An example of CM extraction in one bat is given in Fig 2A.

## Histology

Cochlear histology was assessed in four bats. The bats were euthanized and perfused with PBSx1. The inner ears were dissected, fixed in 4% PFA at 4°C overnight, and decalcified at 4°C for 8 d in a standard decalcifier. The ears were then processed for paraffin sections using the tissue processor (TP1020; Leica). Paraffin blocks were then made using Histo-Embedder (Leica) and sectioned using a microtome (Jung RM2055; Leica). Cochlea mid-turn cuts were taken in all the samples for consistency. Paraffin serial sections (15 $\mu$m) were stained with hematoxylin and eosin using Multistainer (Leica). Slides were imaged using Aperio Slide Scanner (Leica). For quantification of the stria vascularis cross-sectional area, three slides were measured for each ear and averaged using FIJI (ImageJ2). Spiral ganglion neurons and the hair cells of an aged bat (11.3 yr) are shown in Fig S3. Both cell types seem intact, however, because of technical difficulties, we could not reliably quantify these cells' survival.

## Wing sample collection, DNA extraction, and quality control

To determine the bats' age, tissue samples were carefully taken from their wings, avoiding puncturing blood vessels. This procedure has been used many times and does not require anesthetization (e.g., Harten et al, 2018). However, because the bats were already anesthetized for the ABR recordings, the tissue extraction was performed under the anesthetization to avoid unnecessary stress. The tissue samples were taken using 3-mm biopsy (four samples per wing) and kept in absolute ethanol at 4°C until DNA extraction.

Genomic DNA was extracted from the wing biopsies using DNeasy Blood & Tissue Kit (QIAGEN Ltd.) with one modification to the suggested protocol: at the elution stage, 35 $\mu$l of elution buffer was used, in two repetitions, to increase yield. After extraction, all of the samples were quantified using Equalbit dsDNA HS Assay Kit EQ111 (Vazyme Biotech Co., Ltd) using Qubit 2.0 Fluorometer (Invitrogen, Thermo Fisher Scientific Co., Ltd). 30 $\mu$l from samples at a concentration of over 10 ng/$\mu$l was sent to UCLA Neuroscience Genomics Core.

## Age assessment

The protocol used to assess the bats' age was similar to that of Wilkinson et al (2021). Methylation data were generated using the custom Illumina methylation array (HorvathMammalMethylChip40), which tests the methylation levels on 37,492 CpG sites. The DNA samples underwent bisulfite conversion by Zymo EZ DNA Methylation Kit (Zymo Research), Cy3 and Cy5 labeling, hybridization, and scanning (iScan; Illumina).

DNAm levels were determined by calculating the ratio of intensities between methylated and unmethylated sites, and multispecies epigenetic clock that accurately predicts chronological age was used (Wilkinson et al, 2021).

## Noise exposure evaluation

The acoustics in a bats' roost was recorded for 66 h using a calibrated AudioMoth (Open Acoustic Devices). Two AudioMoth devices were placed in crevices where the bats typically perch to record the noise that a bat is exposed to. Maximum levels are a common way to report bats' call intensities (Holderied & Von Helversen, 2003; Hiryu et al, 2007; Surlykke & Kalko, 2008; Jakobsen et al, 2013). The maximum peak was extracted first, and the minimum peak was searched in a 5-ms window following the maximum peak. The noise maximum levels (peak-to-peak) were extracted for each 30-s time bin in two key frequencies (~32 and ~6 kHz), and the percentages of their occurrence were calculated. Night hours (6 h a night) where most of the bats are outside the roost were excluded from the sum of recording hours. Average noise levels were calculated too.

The entire frequency spectrum of the conspecific noise exposure was done using recordings in a laboratory colony with 13 Egyptian fruit bats. A calibrated GRAS 40DP 1/8″ microphone (GRAS Sound & Vibration) was placed 105 cm from the cluster of the bats, and 49 recordings with an average of ~7 s long were taken automatically (over a period of ~2 h), triggered by the bats' vocalizations. The raw recordings were filtered at a range of ±2 kHz for each of the frequencies that were tested in the audiogram (6, 12, 18, 24, 30, and 35 kHz). Maximum peak-to-peak amplitudes were measured then for each of the 49 recordings, and the average was taken as the noise level. The sound levels at 10 cm were estimated based on the physics of sound propagation.

## Statistical analysis

Most of the measurements were statistically analyzed using GLMs fitted in MATLAB. Hearing deterioration rate comparisons between the different frequencies were analyzed by a two-tailed $t$ test with a Bonferroni correction for multiple comparisons.

# Supplementary Information

# Acknowledgements

We thank Dr. Lee Harten for the training on fruit bats' EEG recordings and for bats' capture; Reut Assa and Adi Rachum for their help with handling the bats; Liraz Attia for his help recording bats' vocalizations; Tucker-Davis Technologies (TDT) and Dr. Victor Rush for technical assistance; the UCLA Neuroscience Genomics Core (http://www.semel.ucla.edu/ungc) for methylation data generation; Prof. Steve Horvath and his team for generating age clock estimates; Prof. Michael Charles Liberman for age-related histology consultation; and Naomi Paz for proofreading the article. This research was partially supported by the European Research Council (ERC program behaviorIsland).

## Author Contributions

YC Tarnovsky: data curation, software, formal analysis, investigation, visualization, methodology, and writing—original draft, review, and editing.
S Taiber: data curation, formal analysis, investigation, visualization, methodology, and writing—original draft, review, and editing.

Y Nissan: data curation, methodology, and writing—original draft.
A Boonman: data curation, investigation, and methodology.
Y Assaf: resources, supervision, and writing—original draft.
GS Wilkinson: methodology and writing—original draft.
KB Avraham: resources, methodology, and writing—original draft.
Y Yovel: conceptualization, resources, supervision, funding acquisition, investigation, methodology, project administration, and writing—original draft, review, and editing.

## Conflict of Interest Statement

The authors declare that they have no conflict of interest.

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
