## [Reviewer comments · Life Science Alliance]

Life Science Alliance

Bats experience age-related hearing loss (presbycusis)

Yifat Tarnovsky, Shahar Taiber, Yomiran Nissan, Arjan Boonman, Yaniv Assaf, Gerald Wilkinson, Karen Avraham, and Yossi Yovel

DOI: <https://doi.org/10.26508/lsa.202201847>

Corresponding author(s): Yossi Yovel, Tel Aviv University

Review Timeline:

Submission Date:	2022-11-22
Editorial Decision:	2023-01-05
Revision Received:	2023-02-04
Editorial Decision:	2023-03-06
Revision Received:	2023-03-14
Accepted:	2023-03-15

Transaction Report:

January 5, 2023

Re: Life Science Alliance manuscript #LSA-2022-01847-T

Prof. Yossi Yovel
Tel Aviv University
Tel Aviv
Israel

Dear Dr. Yovel,

Thank you for submitting your manuscript entitled "Bats suffer from age-related hearing loss (presbycusis)" to Life Science Alliance. The manuscript was assessed by expert reviewers, whose comments are appended to this letter. We invite you to submit a revised manuscript addressing the Reviewer comments.

Thank you for this interesting contribution to Life Science Alliance. We are looking forward to receiving your revised manuscript.

Sincerely,

B. MANUSCRIPT ORGANIZATION AND FORMATTING:

Reviewer #1 (Comments to the Authors (Required)):

In this work, Tarnovsky et al. want to determine whether Egyptian fruit bats, echolocating animals naturally exposed to frequent high-intensity sound, are subjected or not to age-related hearing loss like mammals. They assessed the hearing of 47 Egyptian fruit bats population by ABR and CM while estimating their age by methylation profiling.

Based on their results, they suggest that:

- The occurrence of 10 dB age-hearing loss in bats between groups of 2.8 years old average age and 12 years old animals.
- An increased rate of hearing loss with age with higher frequencies
- The defect is both peripheral, with a reduction of ABR and CM thresholds, and central, with an increased latency between central auditory relays.

While this study is solid and provides some important results, some points need to be addressed, corrected, or improved:

Major points:

- The main results about the ABR thresholds at different ages and different frequencies, and the slope analysis should be improved: It is difficult to determine the quality of the slope fit. Please provide statistical analysis to evaluate this fit. Also, I suggest showing, potentially in a supplemental figure, the average threshold for each frequency along with the fit. I understand that the older animals were rarer. Still, one needs to be able to determine how the results from the 2 oldest animals fit compared to the one before because they will drastically influence the fit. While it could occur faster for higher frequencies, the data show a similar magnitude of hearing loss across frequencies, which differs from mouse and human ARHL.
- The analysis of gender biases seems not possible for single or partially grouped ages because of sampling differences:
 - o 2-5 years old: 15 males vs 7 females
 - o 6-8 years old: 3 males vs 14 females
 - o 10-14 years old: 4 males vs 6 females
- It is a missed opportunity not to have analyzed the entire frequency spectrum of the environmental noise exposure of the bats to test if there is a correlation with the rate of age-related HL. The noise intensity contribution per frequency should be plotted as distributions for a better view. Also, what is the hearing range of these animals? Why has a very low frequency not been included?
- How the CM have been recorded is unclear to me. An electrode is usually placed next to the tympanic membrane or into the scala tympani. Please provide a detailed description of the procedure.
- The cochlear histology should be improved: Spiral gg neuron density and the presence of hair cells in the epithelium along the tonotopical axis should be imaged (from histology) and quantified. All results could originate from the degenerescence of hair cells and spiral ganglion neurons.

Minor:

- Different fonts in the sentence : "In addition, we recorded cochlear microphonics (CM) noninvasively, using the same subdermal electrode positioning used for the ABR recordings. The CM response represents activity from the outer hair cells (OHC), which amplify the sound-induced motions in the inner ear, while the inner hair cells (IHC) translate these motions into the chemical signals that excite the auditory nerve (AN) (Liberman, 2015). "
- In significance statement, space missing between <10 gr
- (SPL) Re 20 Pa : what "Re" means?
- what is a GLME?
- Check space between value and unit (ex 8kHz should be 8 kHz)

Reviewer #2 (Comments to the Authors (Required)):

Referee comments on LSA-2022-01847-T

Bats suffer from age-related hearing loss (presbycusis)

Tarnovsky et al.

This is an ambitious study using multiple methods to assess the occurrence and magnitude of age-related hearing loss in Egyptian fruit-bats, *Rousettus*. The use of new DNA methods to assess age in bats, and the use of minimally-invasive neurophysiological recording techniques are useful methodological advances. So is the use of cochlear microphonic recordings.

First, I suggest changing the title and any use of the term, "suffer," to describe presbycusis in *Rousettus*. I suggest using "Bats experience.." instead of "Bats suffer from..." throughout the manuscript. "Suffer" implies punishing effects or debilitating consequences that are not observed, and that brings to mind negative connotations to readers that are not good to evoke. The measured sge-related hearing losses amount to only about 1 dB per year.

Second, I suggest changing "noninvasive" to "minimally-invasive" for the ABR and CM recordings in unanesthetized bats. The Simmons et al 2022 citation makes this distinction desirable.

The absence of page numbers or line numbers in the manuscript make it difficult to specify further suggestions, but the pdf reader does give pages.

Page 4, second paragraph: Linnenschmidt & Wiegrebe, 2019 is not a good general reference to echolocation for readers, so I suggest the two recent Springer books *BioSonar* and *Bat Bioacoustics*.

Page 4, Last paragraph: to the citations of Kick & Simmons, 1984; Liu et al., 2021, add Simmons et al 2016. This reference seems to be the earliest to directly address noise damage effects on hearing and then actually test it behaviorally.

Page 6, first paragraph: cochlear microphonics (CM) seems to be in a different text font than the rest of the paragraph.

Pages 27-28: CM recordings were cleverly done using short clicks and a 30-cm speaker-to-ear tube, plus phase reversals to ensure the recordings were really physiological and not speaker radiation artifacts, which are the most difficult problem for CM recordings.

We thank the reviewers for these comments.

We added page and line numbers.

Changes in the manuscript and the supplementary file are in red letters.

Reviewer #1

Major points:

- The main results about the ABR thresholds at different ages and different frequencies, and the slope analysis should be improved: It is difficult to determine the quality of the slope fit. Please provide statistical analysis to evaluate this fit.

We added the P values of the slopes in the figure's legend (page 8).

Also, I suggest showing, potentially in a supplemental figure, the average threshold for each frequency along with the fit.

We added the thresholds in figure 1D - as blue dots.

I understand that the older animals were rarer. Still, one needs to be able to determine how the results from the 2 oldest animals fit compared to the one before because they will drastically influence the fit. While it could occur faster for higher frequencies, the data show a similar magnitude of hearing loss across frequencies, which differs from mouse and human ARHL.

We are not sure that we understand this comment. First, we note that we did see a higher decrease in hearing at higher frequencies and second, we show all of the thresholds for all bats (of all ages) in figure 1D. We tried to better clarify this in the legend. We apologize if we are missing the reviewer's suggestion.

- The analysis of gender biases seems not possible for single or partially grouped ages because of sampling differences:

- o 2-5 years old: 15 males vs 7 females
- o 6-8 years old: 3 males vs 14 females
- o 10-14 years old: 4 males vs 6 females

We ran a generalized linear model (GLM) with all ages together and found no effect of sex. To our best statistical understanding, this is legitimate as the model takes all individuals into account together. In the revised mns we also examined a model that includes sex-age interaction and found no significant difference in the model' BIC.

- It is a missed opportunity not to have analyzed the entire frequency spectrum of the environmental noise exposure of the bats to test if there is a correlation with the rate of age-related HL. The noise intensity contribution per frequency should be plotted as distributions for a better view. Also, what is the hearing range of these animals? Why has a very low frequency not been included?

Following the reviewer's comment, we estimated noise exposure at all relevant frequencies and examined the correlation with hearing loss. We found an inverse correlation between the rate of age-related hearing loss and noise intensity, with slower deterioration seen for frequencies with more exposure to vocalization noise. Specifically, communication vocalizations were louder at lower frequencies with better hearing thresholds, and these frequencies were less affected by age. We show these results in a new figure 5B and supplementary figure number 1 and we discuss them in the discussion (page 16-17).

The hearing range of this species can be learned from the audiogram presented in Figure 1D. As can be learned from this figure, the frequencies that we used encompasses the main hearing range of adult *Rousettus aegyptiacus*.

- How the CM have been recorded is unclear to me. An electrode is usually placed next to the tympanic membrane or into the scala tympani. Please provide a detailed description of the procedure.

The reviewer is thinking of an invasive measurement of the CM as is typically done in animals, but in humans, non-invasive CM measurements are common. We added more information about the procedure in the Methods (page 25) and added the relevant references.

Our method is similar to the one used in humans in which the CM response is recorded via surface electrodes (Rance et al., 1999; Shi et al., 2012; Sohmer & Pratt, 1976).

Minimally-invasive cochlear microphonics (CM) were recorded here using the same subdermal electrode positioning used for the ABR recordings. Similar recordings using subcutaneous electrodes were used before on cats (Laukli & Mair, 1983) and wistar rats (Pourbakht et al., 2018).

In brief: CM were obtained similarly to ABRs, with some modifications: The bats were presented with a 0.1 msec click that was transferred via a 30 cm long air tube to temporally distinguish between physiological CM and speaker artifacts, and the signals were played separately in condensation and rarefaction polarities (because alternating polarity that is used in regular ABR recordings cancels out the CM).

- The cochlear histology should be improved: Spiral gg neuron density and the presence of hair cells in the epithelium along the tonotopical axis should be imaged (from histology) and quantified. All results could originate from the degenerescence of hair cells and spiral ganglion neurons.

We thank the reviewer for this comment. We have now added a supplementary image (number 3) showing the spiral ganglion neurons and the hair cells of an aged bat (11.3 years). Both cell types seem intact, however, due to technical difficulties, we could not reliably quantify these cells' survival since they require multiple serial slices. Therefore, we cannot exclude a contribution of possible degeneration of hair cells and spiral ganglion neurons to the hearing phenotype of aged bats at this point.

Minor:

-Different fonts in the sentence : : "In addition, we recorded cochlear microphonics (CM) noninvasively, using the same subdermal electrode positioning used for the ABR recordings..."

We fixed the text font.

-In significance statement, space missing between 10[>] gr

-Check space between value and unit (ex 8kHz should be 8 kHz)

We added the space in the significance statement and also checked space between value and unit along the manuscript.

-(SPL) Re 20 Pa : what "Re" means?

Re is for relative, we added it the first time dB SPL was mentioned in the results.

what is a GLME?

We replaced the term GLME with GLM and added “generalized linear model” the first time we mentioned the abbreviation.

Reviewer #2

I suggest changing the title and any use of the term, "suffer," to describe presbycusis in Rousettus. I suggest using "Bats experience.." instead of "Bats suffer from..." throughout the manuscript.

I suggest changing "noninvasive" to "minimally-invasive" for the ABR and CM recordings in unanesthetized bats.

Page 4, second paragraph: Linnenschmidt & Wiegrebe, 2019 is not a good general reference to echolocation for readers, so I suggest the two recent Springer books Biosonar and Bat Bioacoustics.

Page 4, Last paragraph: to the citations of Kick & Simmons, 1984; Liu et al., 2021, add Simmons et al 2016.

We accept and thank the reviewer for these comments, we performed the suggested changes.

Page 6, first paragraph: cochlear microphonics (CM) seems to be a in different text font than the rest of the paragraph.

We fixed the text font

Other changes:

1. We moved the figure showing the recordings from the bats' cave from the supplementary to figure number 5A.
2. We removed the following lines from the discussion as it seemed unnecessary after analyzing the entire frequency spectrum of the conspecific noise and the relation to age-related hearing loss:

“The hearing loss rate at the low frequencies (i.e., 0.87dB per year at 6 kHz), was slightly higher than human-typical levels (between ~0.26 dB and 0.7 dB per year, Kim et al., 2010; Lee et al., 2005; Lohi et al., 2021; Rigters et al., 2018). This might be related to the very intense low-frequency noise that the bats are exposed to in their roost”.

3. We found a minor mistake in the cochlear microphonics analyses. There was no change in the direction of the results, only in a few of the numbers (page 9 lines 193:194, figure 2B, page 26 line 548 and supplementary figure 2).
4. Figures will be uploaded as individual files.

March 6, 2023

RE: Life Science Alliance Manuscript #LSA-2022-01847-TR

Prof. Yossi Yovel
Tel Aviv University
Tel Aviv
Israel

Dear Dr. Yovel,

Thank you for submitting your revised manuscript entitled "Bats experience age-related hearing loss (presbycusis)". We would be happy to publish your paper in Life Science Alliance pending final revisions necessary to meet our formatting guidelines.

- please address Reviewer 2's remaining minor points
- please add a separate figure legends section (with both your main and supplementary figures) to the main manuscript text
- please add ORCID ID for corresponding author-you should have received instructions on how to do so
- please consult our manuscript preparation guidelines <https://www.life-science-alliance.org/manuscript-prep> and make sure your manuscript sections are in the correct order
- please use the [10 author names, et al.] format in your references (i.e. limit the author names to the first 10)
- please add a conflict of interest statement to the main manuscript text

A. FINAL FILES:

B. MANUSCRIPT ORGANIZATION AND FORMATTING:

Sincerely,

Reviewer #1 (Comments to the Authors (Required)):

The authors have addressed all concerns.

Minor points:

Legend Fig1C: acronyms placed on the CM data are not described.

Line 278: what is "ca"?

Line 630: "10 cm" (space missing)

Reviewer #2 (Comments to the Authors (Required)):

The revisions answer my questions

Changes in the manuscript are in red letters.

-please address Reviewer 2's remaining minor points

Legend Fig1C: acronyms placed on the CM data are not described.

We added the following description: It is commonly accepted that wave I represents the compound response from the auditory nerve (AN), while the later waves represent responses from the ascending auditory pathway: the cochlear nucleus (CN), the superior olivary complex (SOC), the nucleus of the lateral lemniscus (NLL), and the inferior colliculus (IC).

Line 278: what is "ca"? -**We replaced “ca.” with “approximately”**

Line 630: "10 cm" (space missing)- **Space was added.**

-please add a separate figure legends section (with both your main and supplementary figures) to the main manuscript text

We added a separate figure legends section to the main manuscript text (pages 46-47).

-please add ORCID ID for corresponding author-you should have received instructions on how to do so

We added ORCID ID for the corresponding author (page 1)-

<https://orcid.org/0000-0001-5429-9245>

-please consult our manuscript preparation guidelines <https://www.life-science-alliance.org/manuscript-prep> and make sure your manuscript sections are in the correct order

We made few order changes (pages 1 and 30).

-please use the [10 author names, et al.] format in your references (i.e. limit the author names to the first 10)

We edited our references using the [10 author names, et al.] format.

-please add a conflict of interest statement to the main manuscript text

We added a conflict of interest statement to the main manuscript text (page 30)

March 14, 2023

RE: Life Science Alliance Manuscript #LSA-2022-01847-TRR

Prof. Yossi Yovel
Tel Aviv University
Tel Aviv
Israel

Dear Dr. Yovel,

Thank you for submitting your Research Article entitled "Bats experience age-related hearing loss (presbycusis)". It is a pleasure to let you know that your manuscript is now accepted for publication in Life Science Alliance. Congratulations on this interesting work.

DISTRIBUTION OF MATERIALS:

Again, congratulations on a very nice paper. I hope you found the review process to be constructive and are pleased with how the manuscript was handled editorially. We look forward to future exciting submissions from your lab.

Sincerely,
